# Association of the Gut Microbiota with Weight-Loss Response within a Retail Weight-Management Program

**DOI:** 10.3390/microorganisms8081246

**Published:** 2020-08-16

**Authors:** Samitinjaya Dhakal, Lacey McCormack, Moul Dey

**Affiliations:** Department of Health and Nutritional Sciences, Box 2275A South Dakota State University, Brookings, SD 57007, USA; Samitinjaya.Dhakal@jacks.sdstate.edu (S.D.); Lacey.McCormack@sdstate.edu (L.M.)

**Keywords:** gut microbiota, weight-loss, energy restriction, body composition

## Abstract

Retail programs offer popular weight-loss options amid the ongoing obesity crisis. However, research on weight-loss outcomes within such programs is limited. This prospective-cohort observational study enrolled 58 men and women between ages 20 and 72 years from a retail program to assess the influence of client features on energy-restriction induced weight-loss response. DESeq2 in R-studio, a linear regression model adjusting for significantly correlating covariates, and Wilcoxon signed-rank and Kruskal–Wallis for within- and between-group differences, respectively, were used for data analyses. An average 10% (~10 kg) reduction in baseline-weight along with lower total-, android-, gynoid-, and android:gynoid-fat were observed at Week 12 (all, *p* < 0.05). Fifty percent of participants experienced a higher response, losing an average of 14.5 kg compared to 5.9 kg in the remaining low-response group (*p <* 0.0001). Hemoglobin-A1C (*p* = 0.005) and heart rate (*p* = 0.079) reduced in the high-response group only. Fat mass and A1C correlated when individuals had high android:gynoid fat (*r* = 0.55, *p* = 0.008). Gut-microbial *β*-diversity was associated with BMI, body fat%, and android-fat (all, *p* < 0.05). Microbiota of the high-response group had a higher baseline OTU-richness (*p* = 0.02) as well as differential abundance and/or associations with *B. eggerthi, A. muciniphila, Turicibacter*, *Prevotella,* and *Christensenella* (all, *p*/*p_adj_* < 0.005). These results show that intestinal microbiota as well as sex and body composition differences may contribute to variable weight-loss response. This highlights the importance of various client features in the context of real-world weight control efforts.

## 1. Introduction

The burgeoning field of microbiota has revealed how inner environment can shape human health and wellbeing. Gut bacterial compositions may also contribute to and help predict physiological dysfunctions in patients [1,2,3]. In particular, pre-clinical and clinical studies have reported links between gut bacteria and obesity [4,5,6,7]. Experiments in germ-free mice established a direct contribution of the gut bacteria composition and metabolism on body weight outcomes [8,9]. Microbiomes in obese mice and obese humans showed enhanced energy harvest capacity from non-digestible carbohydrates by way of short chain fatty acid (SCFA) production. Furthermore, germ-free mice and rats with low SCFA-levels are protected from diet-induced obesity. Paradoxically, SCFA-intervention protected mice from diet-induced obesity. It is also known that SCFAs modulate fatty acid synthesis and oxidation to favorably impact body fat and weight. However, molecular mechanisms underlying the apparent paradox related to SCFAs as well as other aspects of microbiota mediated host energy metabolism are not fully understood [8,10,11,12,13,14].

In parallel to mechanistic research endeavors, there is a growing interest in the manipulation potential of the gut microbiota for achieving effective weight-loss. While it was proposed that microbiota features may interplay with weight response to interventions, human data on this interplay are limited under real-life circumstances. Furthermore, given the general complexity of the microbiome structure and function, consensus is lacking at present regarding specific microbial compositional features that may help predict weight-loss response outcomes [15,16,17,18,19]. The microbiome composition is unique to each person and is relatively resilient during adult life. However, its responsiveness to environmental cues provides the opportunity to explore its potential role in understanding weight-loss response within real-life settings [4,6,7,15].

Two in three Americans are overweight or obese due to excess body-fat, which elevates the risk of life altering chronic diseases. Complications arising from excess body-fat and body-weight account for over 200 billion dollars in direct annual healthcare cost in the United States [20,21]. Various weight-loss interventions are widely available including many diets, physical activities, behavior therapies, pharmacological treatments, and bariatric surgery. However, according to a National Institutes of Health workgroup, losing weight and preventing weight-regain continue to be challenging for many [20,22]. Assessment of contributing factors to such challenge may help facilitate better-effective weight control measures.

Lifestyle modification is the primary means for achieving and maintaining a healthy body weight. It may include some combination of energy restriction, physical activity, and behavioral adaptations [23]. Guidelines are available to direct overweight and obese patients to participate in community-based lifestyle intervention programs [24,25]. However, the response of participants may vary within and between such programs [26,27]. This underscores the need for increased research looking into factors that may impact consistent outcomes in weight-management programs.

Retail programs offer lifestyle intervention options for weight management on a for-profit basis. Some may offer customized weight maintenance strategies tailored to patient’s need and ability. Their ubiquitous presence within communities across western nations provides clinicians and employers convenient referral options [24]. In a trial, patients referred to a commercial weight-loss program lost twice as much weight as patients who only received weight counselling within the clinical setting. The researchers proposed that due to healthcare related priorities beyond weight-control, physicians typically lack the time needed for weight-loss counselling compared to retail programs that are dedicated to weight-loss solutions [28] Thus, the trial data highlight a potentially important role for reputable commercial programs in organized obesity mitigation efforts. However, high dropout rates may be common and can negatively impact overall effectiveness of even well-established commercial programs [29].

In this study, we examined associations of gut microbiota with 12-week changes in body weight, body fat composition, and distribution, as well as whether baseline microbiota profile, age, or sex may contribute to planned weight-loss response. We worked with a retail program in the local community to recruit study participants. With 168 retail locations across the nation, the program uses one-on-one coaching by trained health-coaches to help adapt to an energy-restricted diet plan that includes formula diet for the first several months into the program. The program recommends maintaining the habitual daily activity level for individual participants while being on the program.

## 2. Materials and Methods

### 2.1. Study Design, Setting, and Participants

Institutional Review Board for Human Subject Research at South Dakota State University approved (IRB-1608010-EXP) this prospective cohort, 12-week, observational study. Incoming clients in a weight-reduction program from its Brookings county retail location in South Dakota were recruited through flyers between November 2016 and February 2018. Fourteen men and 44 women—between ages 20 and 72 years and predominantly Caucasian due to geographical location—joined the study on a first-come-first-serve basis after going through eligibility screening and providing informed consent in writing. Despite our best effort, a low number of males signed up for the study. This experience is similar to another study reporting that men are less likely than women to participate in retail weight-loss programs [30]. Inclusion criteria included being generally healthy (i.e., absence of major illnesses including diabetes), any ethnicity and either sex, willing to undergo planned weight-loss by adhering to the guidelines of the retail-program, and willingness to comply with the research study protocol. Exclusion criteria were pregnancy; lactation; probiotic-, prebiotic-, and antibiotic-usage; being on any special diet in the previous 12-months; and cancer, diabetes, immune compromised state, and other conditions that would affect the ability to provide informed consent or comply with the study protocol. Nineteen out of 58 recruited participants (32.8%) dropped out of the program before Week 12 and were excluded from the study. There was no case where a participant continued on the retail program but discontinued with the study (Figure 1). This led to a longer than anticipated recruitment phase so that sufficient number of participants could still complete the study to allow 85% probability of detecting a weight-loss at a 0.05 significance level. Of note, as high as 58% dropout by 13th week into a well-known commercial weight-loss program has been previously reported by other researchers [29].

### 2.2. Diet, Anthropometrics, and Other Data Collection during Participant Visits

All data and samples were collected at two time points: Week 0 (baseline), i.e., right after enrollment in the program, but before starting the plan-diet, and at Week 12, i.e., after continuing to follow the program-diet for 12 weeks. The 12-week time period was previously observed as an optimal average time window when majority of program participants tend to be within 50% of their target weight (personal communication from McCormack-group, Brookings, SD, USA). Furthermore, clients were required to consume pre-formulated, ready-to-eat food items sold by the program during the initial months for maximizing weight-loss prior to transitioning to regular food. This provided greater uniformity in dietary ingredients and patterns among participants at Week 12 data collection, potentially enhancing quality and consistency of the reported outcomes. Dietary-intake data were collected in-person by nutrition science and dietetics students under supervision of a registered dietitian nutritionist using 24-h recalls and were entered in the Nutritionist Pro software connected to USDA nutrition information database (Axxya Systems, Redmond, WA, USA) [31]. Recall data were collected during the scheduled study-visits for two non-consecutive days (specific days for an individual were decided during the interview) from recent past of which one was a weekday and the other a weekend-day. Each participant was provided with a wallet-size WebMD portion size guide for reference (2012 version by Kathleen Zelman, MPH, RD, LD). For packaged food items, nutrition data were obtained from labels.

Height, waist (at navel), and hip (at the widest part) were recorded to the nearest 0.5 cm while body weight (BW) was measured to the nearest 0.1 kg in light clothes or scrubs and no shoes (electronic scale, Seca GmbH & Co., Hamburg, Germany) following established protocols [32]. Waist:hip (WH) and body mass index (BMI) are two widely used markers in obesity control programs due to simplicity of measurements and low-cost tools. However, since these were reported to be less useful for metabolic risk assessment [33,34,35], we additionally assessed body composition features using dual energy X-ray absorptiometry (DXA) whole body scan (Hologic QDR Discovery, Waltham, MA, USA) that included—total fat mass, body fat percentage (BF%), android fat, gynoid fat, and amount and distribution of corresponding fat-free masses (to indicate lean masses) [36,37,38,39]. In presenting WH, BMI, and body composition data side-by-side, we pointed out when one measure may not serve as a surrogate for the other.

Glycated hemoglobin (A1C), an indicator of glucose metabolism and diabetic status, was measured using A1C Now^®^ + kit (PTS Diagnostics, Indianapolis, IN). Individuals with an A1C of ≥6.5% were considered diabetic range and excluded from the study [40]. The retail plan is administered differently to diabetics which could have potentially introduced variability in response data. Blood pressures and resting heart rates were measured using a digital sphygmomanometer (Greater Goods, LLC, St. Louis, MO, USA) using standard procedures [32].

### 2.3. Nucleic Acid Extraction and 16S Ribosomal RNA Amplicon Sequencing

Participants were provided with collection hat, gloves, and Omni-Gut stool collection kit (DNA Genotek, Ontario, ON, Canada). The samples were received within 24 h after collection, processed, aliquoted, and stored at −80 °C. DNA was isolated with MagAttract PowerMicrobiome Kit following manufacturer’s guidelines (Qiagen, Valencia, CA, USA) and quantified with Qubit^®^ Quant-iT High Sensitivity Kit (Invitrogen, Life Technologies, Grand Island, NY, USA). 16S V4 rRNA region was enriched using PCR amplification using primers designed against the surrounding conserved regions, followed by ligation of adapters and indexing barcodes (Illumina, San Diego, CA, USA). PCR products were quantified by PicoGreen (Life Technologies, Grand Island, NY, USA) and prepared equimolar for sequencing step.

Sequencing (250 cycles, paired-end) using Illumina Miseq platform (San Diego, CA, USA) was carried out by Second Genome (South San Francisco, CA, USA). Sequenced reads were quality filtered and resulting unique sequences were clustered at 97% by UPARSE (de novo OTU clustering) and a representative consensus sequence per de novo OTU was determined. The UPARSE clustering algorithm comprises a chimera filtering and discards likely chimeric OTUs. Resulting sequences were searched against Greengenes reference database (closed reference operational taxonomic unit (OTU) picking). The longest sequence from each OTU was then assigned taxonomic classification via Mothur’s Bayesian classifier, trained against the Greengenes database clustered at 99%. Further bioinformatics and data analyses (except Piphillin) were carried out in our laboratory (next sections). We also carried out BLAST search to cross reference with closest hits from NCBI 16S rRNA database with >95% query cover, >87% identity, and <0.01 E value for unidentified OTUs in Greengenes. The raw sequences are deposited in NCBI sequence read archive (SRA, accession number SRP237387), belonging to BioProject number PRJNA595387.

### 2.4. Sub-Grouping of Participants for Data Analyses

All data are presented for 36 participants (ALL) as well as after grouping them based on baseline differences in age, sex, BMI, and android:gynoid fat ratio (AG) to assess any role of these participant characteristics in weight-loss. Participants are also grouped based on the extent of body-weight reduction (weight-loss response) at Week 12. Data are presented as: within group differences from baseline to Week 12 and between group differences at baseline and Week 12. The subgroups are defined as follows: (1) young adults (YA, *n* = 20, average age 33 years), and older adults (OA, *n* = 16, average age 62 years; (2) males (*n* = 9) and females (*n* = 27); (3) higher BMI (HI_BMI_, *n* = 24, average BMI 38 kg/m^2^) and lower BMI (LO_BMI_, *n* = 12, average BMI 27 kg/m^2^), where the BMI cut off was determined based on normal + overweight (<29.9 kg/m^2^) and obese ranges (>30 kg/m^2^); (4) higher AG (HI_AG_, *n* = 23, average AG 0.63) and lower AG (LO_AG_, *n* = 13, average AG 0.43), where lower AG is generally deemed healthier, but, unlike BMI, currently there is no reference value available, thus the arbitrary cut-off for our mixed sex cohort was taken as 0.5, partially based on a study that reported 50th percentile of AG for 50 years and older Caucasians as 0.46 (female) and 0.71 (male) [39]; and (5) the higher response (HI_res_, *n* = 18) and lower response (LO_res_, *n* = 18) groups were based on a 10 kg weight-loss cut off. Significant difference in the respective defining criterion between the subgroups was present for: age (YA/OA), BMI (HI_BMI_/LO_BMI_), AG (HI_AG_/LO_AG_), and weight-loss response (HI_res_/LO_res_).

### 2.5. Bioinformatics and Statistical Data Analyses

All data analyses were carried out using R-studio and/or Sigma Plot Software (Systat Software Inc., San Jose, CA, USA) and data presented as mean ± SD unless otherwise stated. Normality was assessed using Shapiro–Wilk test followed by *t*-test or Mann–Whitney U-test for analysis of anthropometric and metabolic features. Statistical significance was considered at *p* ≤ 0.05. A *p* value greater than 0.05 and less than 0.08, when shown, indicates approaching significance.

For metagenomics data, multiplexed sequence reads were converted to taxonomic and phylogenetic profiles using QIIME2 (Quantitative Insights Into Microbial Ecology). Alpha diversity i.e., OTU richness and Shannon diversity were calculated by summing unique OTUs found in each sample and combining richness with the relative abundance data, respectively. For beta diversity, dissimilarity score was determined by comparing in a pairwise fashion using a distance matrix. Abundance-weighted difference was calculated using Bray–Curtis and binary dissimilarity were identified using Jaccard index [41,42]. Statistical testing of alpha-diversity was carried out using Linear Regression, Wilcoxon, and Kruskal–Wallis tests. Permutational Analysis of Variance (PERMANOVA) testing utilized the sample-to-sample distance matrix directly, and not a derived ordination or clustering outcome, to find the differences among discrete, categorical, or continuous variables by randomly reassigning the samples to various sample categories. Correlations were determined using Pearson’s coefficient. Univariate differential abundance of OTUs was calculated using DEseq2 package with default settings in R-studio [43]. The package uses a negative binomial noise model for the overdispersion and Poisson process intrinsic to the data [44]. Additionally, the package also takes both technical and biological variability between experimental conditions into account and provides the difference in terms of Log_2_ fold-change. Furthermore, the *p* value was adjusted (*p_adj_*), when appropriate, using Benjamini–Hochberg correction for false discovery rate (FDR) inherent to large number of dependent variables [45]. The functional capacity of the metagenome from OTU counts and representative sequence of each OTU was assessed using Piphillin version 6.0 (Second Genome Inc., South San Francisco, CA, USA). The closest matched 16S rRNA sequence above the identity cut-off at 95% was considered as the inferred genome for that OTU. In the case of multiple nearest neighbor genomes with equal identities, the count is equally split to sum the inferred genome content. The content is expressed as orthologs searched against pathways counts in Kyoto Encyclopedia of Genes and Genomes (KEGG) genome database [46].

## 3. Results

### 3.1. Energy-Intake in Study Participants

At Week 12, there was an average 21.8% lower daily calorie intake than at baseline in ALL (Table 1, *p* = 0.000003). Energy intake from carbohydrates was lower while that from proteins was higher at Week 12 (both, *p* ≤ 0.01). Average fiber intake—critical for nourishing the gut microbiota—did not change between the microbial data collection time points in the study, i.e., from baseline to Week 12. Intakes were similar between HI_res_ and LO_res_ at baseline and at Week 12 (all, *p* > 0.05). Furthermore, HI_res_ and LO_res_ experienced a similar average reduction of 531.7 and 485.8 kcal/day, respectively, over the 12-week period (*p* > 0.05).

### 3.2. Baseline Body Composition: BMI versus AG

A growing body of research points to BF%, android region fat, and AG (relative measure of android fat) being more relevant than frequently utilized measures of BW and BMI for assessing chronic disease risk [47,48,49,50,51,52,53,54,55]. Overall, two participants had BMI < 25 kg/m^2^ at baseline, while the rest were overweight, obese, or severely obese (BMI range 24.42–51.84 kg/m^2^). Males started with an average 19 kg higher BW than females (*p* = 0.04), although the BMI was similar between the groups (Table 2). However, males had a higher AG than females (*p* = 0.001). In addition, total body fat was 10% lower, while fat-free mass was higher by approximately 20 kg (both, *p* < 0.001) in males than females (Table 2). The observations support that BMI, but not AG, may be influenced by lean-mass and/or height. Thus, AG may be more relevant than BMI in deciding weight-reduction goals. Furthermore, measures of metabolic functions such as A1C (*p* = 0.014) as well as systolic and diastolic blood pressures (both, *p* < 0.001) were higher in the HI_AG_ than the LO_AG_ group, but this distinction was absent between the BMI subgroups (Table 2). These observations indicate that AG is superior to BMI for indicating body fat content and distribution as well as for assessing possible metabolic implications of increased fat in the central region of the body. Of note, the average age difference of 12 years was significant between the AG groups (*p* = 0.025) but such a distinction was not observed between the BMI groups. Central obesity tends to increase with age when overall metabolism tends to slow down. This may explain higher age of the HI_AG_ group (Table 2). Our observations are in line with previous reports that body fat and its distribution pattern independently associate with higher risk of chronic disease and mortality [34,35,53,54]. When possible, retail programs may consider body composition measurement as an option for determining weight-loss and metabolic wellbeing goals.

### 3.3. Anthropometric and Metabolic Changes over 12-weeks

Seventeen participants lost ≥10%, fifteen participants lost 5–10%, three participants lost ≤5%, and one participant gained approximately 2% of their respective baseline BW. Average weight lost in 12 weeks was ~10 kg (Table 2 and Table 3) and the BW reduction was significant in all sub-groups (Table 3 and Table 4). Along with body-weight, the following measures were reduced: blood pressure, total fat and lean masses, BF%, android fat and lean masses, android fat%, gynoid fat and lean masses, and AG (all, *p* ≤ 0.05, Table 4). An association between reduction in fat mass and A1C (*r* = 0.49, *p* = 0.0028) was observed in ALL that was further augmented in HI_BMI_ and HI_AG_ (both, *r* ≥ 0.55, *p* < 0.01) but absent (*p* > 0.3) in corresponding low-groups (Figure 2). There was no apparent relationship between participant’s age and how they responded to energy reduction for weight-loss (Table 2 and Table 3).

### 3.4. Response Variability in Anthropometric and Metabolic Outcomes

Influence of sex: Males experienced an overall higher weight-loss than females as reflected by greater decrease in: BW and BMI (both, 12.9% versus 9%, *p* = 0.031), total fat mass (24.1% versus 13.9%, *p* = 0.001), BF% (13.3% versus 5.7%, *p* = 0.001), android fat mass (32.47% versus 18.5%, *p* = 0.002), gynoid fat mass (23.2% versus 14.2%, *p* = 0.007), AG (13.1% versus 5%, *p* = 0.02), and heart rate (11% decrease versus a small increase, *p* = 0.05) (Table 3 and Table 4). Although the sample size for males was smaller in the study, higher weight-loss in males than females align with previous reports [56,57]. Of note, a previous study reported that men are less likely than women to participate in retail weight-loss programs [30].

Response variability not explained by anthropometric characteristics: Exactly half of the participants experienced a higher degree of weight and fat loss than the rest at Week 12—an average reduction of 13.2% (14.5 kg) versus 6.8% (5.9 kg) of baseline BW; this corresponded to an average 10% versus 5.2% lower BF% (both, *p* < 0.05) at Week 12 than baseline in HI_res_ and LO_res_, respectively. In addition, HI_res_ had a greater (6.5%, *p* = 0.005) reduction in A1C than LO_res_ (3.2%, *p* = 0.072) at Week 12 (Table 4). However, with the exception of a higher starting BW in HI_res_ (*p* = 0.003), the two response-groups had similar age, body-composition types (BF% and AG), A1C, blood pressures, and heart rates at baseline (all, *p* > 0.05, Table 2). Notably, changes in energy intake over the 12-week period were similar between the two response groups (Table 1).

### 3.5. OTU Richness May Influence Weight-Loss Response

The number of sequences and total number of OTUs after independent filtering were ~9.8 million and 1133, respectively. The most abundant microbial phyla at baseline were Firmicutes (68.6%), Bacteroidetes (23.7%), Actinobacteria (4.04%), Proteobacteria (1.79%), Verrucomicrobia (1.36%), Euryarchaeota (0.26%), Tenericutes (0.13%) and Cyanobacteria (0.09%). The most abundant microbial families were Lachnospiraceae (28.4%), Ruminococcaceae (26.5%), Bacteroidaceae (17.1%), Coriobacteriaceae (2.25%), Erysipelotrichaceae (2.12%), Rikenellaceae (2.04%), and Porphyromonadaceae (1.67%).

Mean OTU richness increased from 512 at baseline to 543 at Week 12 in ALL (*p* = 0.017) (Figure 3). The two groups who lost most weight, i.e., most responsive to weight-loss, started with significantly higher OTU richness at baseline—HI_res_ (551) and males (567)—than LO_res_ (473, *p* = 0.02) and females (493, *p* = 0.03), respectively. Post weight-loss, the richness in the HI_res_ (566) and males (556) remained more or less unchanged compared to large increases in LO_res_ (521) and females (539), respectively. Four response groups were similar in age (Table 2). The observations supports a previous report that a greater existing alpha diversity predicts an increased resistance to further diversification of the microbiome [58]. Shannon diversity ranged from 4.09 to 4.15 across all samples and did not differ significantly between sub-groups or from baseline to Week 12.

### 3.6. Changes in Beta Diversity

Samples from same subject clustered together with strong inter-individual differences. Abundance of Firmicutes (*p* = 0.052) and Actinobacteria (*p* = 0.03) decreased while that of Tenericutes (*p* = 0.06) and Euryarchaeota (*p* = 0.05) as well as that of family Porphyromonadaceae (*p* = 0.00011) increased at Week 12, in ALL. PERMANOVA of distance matrices showed general association of beta diversity with age, heart rate, total fat mass, BF%, android fat mass and android fat% (all *p* < 0.05), indicating contribution of these factors to the overall microbiome differences among individuals. These relations did not change from baseline to Week 12 and, hence, were likely not influenced by weight-loss or the calorie-restricted diet. In ALL, *Turicibacter sp*. (*r* = −0.55, *p* = 0.001) and *Christensenella sp*. (*r* = −0.61, *p* = 0.012) negatively correlated with BF% at baseline but not at Week 12. *Christensenella minuta*—the first identified species in this genus—was previously associated with healthy, low-BMI individuals and short chain fatty acid production in previous studies from our and other groups [59,60].

### 3.7. Microbiome Differences in Response Groups

*Prevotella copri* increased in HI_res_ but decreased in LO_res_ from baseline to Week 12 (both *p_adj_* < 0.014). Two OTUs of *Ruminococcus* were differentially abundant between HI_res_ and LO_res_ groups at baseline only (both, *p_adj_* < 0.005, Table 5). One OTU of *Akkermansia muciniphila* showed more than 10-fold higher abundance in LO_res_ compared to HI_res_ consistently at baseline and at Week 12 (both, *p_adj_* < 0.003). In addition, an OTU of *Bacteriodes eggerthi* showed over 50-fold higher abundance at baseline in HI_res_ than LO_res_ (*p_adj_* = 0.005) and associated with weight- and BMI-change in HI_res_ (*r* = 0.60, *p* = 0.008) but not in LO_res_. Abundance of *Bacteriodes plebius* and *Eubacterium biforme* were consistently lower in HI_res_ and males at baseline (all, *p_adj_* ≤ 0.03). Interestingly, unlike various differences in individual OTU-level response between HI_res_ and LO_res_, fewer distinctions were observed between males and females over the 12-week period (Table 5 and Table 6). The data from HI_res_ and LO_res_ show potential contribution of microbiome differences to weight-loss variations. However, sex may influence weight-loss in a manner that may or may not be linked to microbiome differences.

### 3.8. Microbiome Associations in Age-Groups

Associations observed throughout the study: Although age differences did not appear to influence weight-loss response, differential age-specific microbial associations were observed. OTUs belonging to *Prevotella*, and *Paraprevotella* genera as well as to species of *Bacteriodes plebeius* and *Eubacterium biforme* showed higher relative abundance in YA than OA at both time points indicating potential age-related association (all, *p_adj_* ≤ 0.04, Table 6). *Prevotella* (*r* = −0.53, *p* = 0.03 at baseline, and *r* = −0.63, *p* = 0.01 at Week 12) and *Eubacterium biforme* (*r* = −0.54, *p* = 0.069 at baseline and Week 12) had inverse correlations with android fat% in OA (Table 7). *Coprobacillus* positively correlated with AG (*r* = 0.54, *p* = 0.031), WH (*r* = 0.70, *p* = 0.002) in OA.

Time-point specific observation: At baseline, *Christensenella sp.* negatively correlated with BF% (*r* = -0.76, *p* = 0.011), *Parabacteroides distasonis* with A1C (*r* = −0.55, *p* = 0.027), and *Prevotella* (*r* = −0.63, *p* = 0.009) and *Anaerotruncus* (*r* = −0.50, *p* = 0.048) with android fat%, in OA. On the other hand, *Lactobacillus* (*r* = 0.67, *p* = 0.001) and *Bifidobacterium* (*r* = 0.51, *p* = 0.022) associated with BMI in YA. Species of *Lactobacillus* and *Bifidobacterium* are used as probiotics and consumed for health promotion, although the evidence on their effect on weight-loss has been conflicting [61,62]. *Turicibacter* showed an inverse association with BF% (*r* = −0.66, *p* = 0.002) and gynoid fat mass (*r* = 0.51, *p* = 0.026) and *Ruminococcus torques* with BF%(*r* = −0.52, *p* = 0.019) in YA (Table 7). At Week 12, *Dorea formicigenerans* negatively correlated with AG in OA (*r* = −0.62, *p* = 0.01). In YA, a species of *Anaerostipes* showed a negative correlation with BMI (*r* = −0.51, *p* = 0.062), BF% (*r* = −0.59, *p* = 0.026), total fat mass (*r* = −0.58, *p* = 0.03), and gynoid fat mass (*r* = −0.54, *p* = 0.046).

### 3.9. Predicted Metabolic Functions

Functional inference of the metagenomic shifts between baseline and Week 12 was carried out [46]. The Piphillin-generated hits against KEGG Orthologs (KO) abundance data showed potential modulation of 297 functional pathways based on gene content predictions for protein-coding genes. Several pathways showed subtle but statistically significant changes in KO abundance at Week 12 in ALL (Table 8). This included lower KO abundance for starch and sucrose metabolism as well as primary and secondary bile acids (all, *p* < 0.05). Notably, participants consumed fewer dietary carbohydrates at Week 12 compared to baseline (Table 1).

This section may be divided by subheadings. It should provide a concise and precise description of the experimental results, their interpretation, and the experimental conclusions that can be drawn.

## 4. Discussion

National guidelines for weight-loss and chronic disease prevention recognize that referral outside of the clinical setting to weight-loss programs, including retail programs, is needed for effective weight management and improved public health [24]. However, reports of client experience in terms of weight-loss outcomes in commercial real-life settings are relatively scarce. As obesity rates continue to rise, it is critical to dedicate research into factors relating to effectiveness of such programs. This study is among the first to present a detailed evaluation of gut microbiota features in the context of weight-loss response-outcome within a commercial weight-management program. Simultaneous evaluation of the gut microbiota and an extensive array of body composition parameters helped assess the relationship of these outcomes with weight-change response in these participants. The main observations that emerged are: (1) All but one participant achieved significant weight-loss over 12 weeks. However, the extent of weight-loss was non-uniform; half of them lost more than twice as much body weight as the rest despite a similar level of energy restriction and same formula diet between the response groups. Individuals with higher response had higher OTU richness (alpha diversity measure) and taxa level differences (beta diversity measures) compared to the low response individuals. The results indicate that differences in the gut microbiota of participants may relate to the response to weight-management interventions. (2) A1C associated with fat mass and was attenuated in the high-response group but not in the low-response group. It is possible that a larger weight-loss (>10% of initial weight in this cohort) may be necessary when a significant A1C reduction is desired such as in type 2 diabetics. Ironically, losing weight is particularly challenging for individuals with type 2 diabetes [63]. (3) Aging is associated with metabolic alterations that can lead to excess weight gain and slower weight-loss [64,65]. Thus, we hypothesized that participant’s age may influence weight-loss response. However, we did not observe a differential weight-change between young and older adults.

Adherence of participants is one of the many factors that may contribute to the impact of a program on community health and fitness. A low initial weight-loss may reduce motivation to adhere in low-responders and can lead to up to 27% reported dropout rate by end of first four weeks [29,66]. The observed extent of variability in weight-loss response in as many as half of our participants even after 12 weeks was striking from that context. As a next step, our results indicate a potential relationship between baseline gut microbiota and the variability in weight-loss response. Educating incumbent program participants regarding potential response-variability due to their own baseline gut features may help set realistic expectations and improve adherence. Furthermore, observational research such as this contributes to the groundwork needed for mechanistic research on how to precisely predict response-level in patients based on their gut signature. We anticipate that, with increased application of microbiome research knowledge in practice, routine metagenomics profiling in retail participants would be logical in the future. On a side note, the high cost of retail programs could also lead to low-adherence [24]. A third of the recruited participants chose to discontinue the retail program and were dropped from the study. Common reasons for quitting were: (i) high cost of proprietary food items; and (ii) dislike for low-calorie formula diet. In contrast, reasons to continue were: (i) convenience of ready-to-eat foods; and (ii) reinforcement through personalized coaching. While quantitative assessments of these responses were beyond the scope of this research, they align with a prior report of even higher dropout (58%) from another well-established commercial program due to costs and other barriers [24,29]. Therefore, validation of strategies to facilitate adherence must be a priority.

The presented metagenomics observations support the increasingly recognized contribution of the gut microbiome toward human health, metabolism, and treatment response [7,67]. *B. eggerthii* was shown to correlate with body fat and inflammation [68,69]. This corroborates with our observation of higher abundance of this species in the high-response individuals who had higher baseline android fat. However, we also observed a hitherto unknown direct association of *B. eggerthii* with weight-reduction selectively in high-responders. Whether higher abundance of this species on the one hand indicates increased body fat, but on the other hand enhances responsiveness to energy-reduction warrants follow-up research. Further, *Christensenella minuta* has been linked with healthy body composition and *Turicibacter* with reduction in high-fat diet induced inflammation [59,60,69,70]. Our data showing negative correlation of *Christensenella* and *Turicibacter* with body-fat while undergoing weight-loss support these prior observations.

A favorable role of *Prevotella* abundance on weight-loss, especially in the context of a high fiber diet, has been reported [71]. We observed an inverse association of the *Prevotella* genus with android fat post weight-loss in older individuals, and an increased abundance of *Prevotella copri* in high-response group. Our results also indicate an overall metabolic shift post weight-loss that included changes in carbohydrate metabolism. More recently, it was proposed that *Prevotella* enterotype patients with genetically low starch digestion capability may produce higher levels of SCFA that contributes to greater weight-loss [72]. The concept of microbial enterotypes attempts to simplify patient stratification based on complex gut signature, although consensus is lacking among researchers regarding the extent of its usefulness [73,74,75,76]. Nevertheless, our observations along with prior evidence direct to the possibility that SCFA-producing *Prevotella* spp. may at least partially drive the metabolic shift needed for weight-loss. Furthermore, for additional insights on bacteria derived metabolites (e.g., SCFAs) associated with weight-loss response, fecal metabolomics profiling may be considered in the future.

Dao et al. reported association of *A. muciniphila* with a healthier metabolic status in calorie-restricted overweight and obese adults [18]. We observed a higher abundance of this species in low responders who had lower BMI than high responders. A lower BMI suggests a healthier metabolic status. However, Dao et al. reported no difference in weight-loss between high and low baseline *A. muciniphila* groups. Another trial reported no reduction in visceral fat and BMI after three months of *A. muciniphila* supplementation [77]. Therefore, it is possible that the observed differential *A. muciniphila* levels in our response groups is more indicative of the differences in their metabolic status than the differential response to weight-loss. A larger body of independent research would help determine the precise role of *A. muciniphila* in weight-loss versus maintenance of healthier metabolism.

The study is not without limitations: (a) This is not an intervention/mechanistic study and does not establish whether the observed relationships are causal or consequential in nature. However, being among the first human studies presenting the relationship of the gut microbiome with weight-loss response within a real-life retail program setting, this exploratory research contributes to the groundwork needed for larger observational as well as mechanistic research. (b) Our observational and small study cohort, although adequate from statistical perspective, was racially homogeneous. It mostly represented Caucasians owing to the geographical location of the study site. The findings may not directly apply to other races as host-genetics may influence microbiome characteristics [78]. (c) Species in the mucus layer (e.g., *A. muciniphila*) differ from those in the intestinal lumen. Stool samples—a practical and non-invasive way to assess gut microbiota in human subjects—represent more luminal species and may not fully reflect the entire gut microbiota [79]. (d) Since the retail program has everyone undergoing energy restriction, an eucaloric control arm was not feasible. However, we argue that the response sub-groups consumed same diet and underwent similar level of calorie-restriction; therefore, differences in weight-loss outcomes are likely contributed by client-features such as the gut microbiota profile. Furthermore, we believe that understanding real-life outcomes despite logistical challenges has its merits.

## 5. Conclusions

With the obesity epidemic showing no sign of decline, it is increasingly recognized that retail weight-management programs can play important role in mitigation efforts. We examined the extent of weight-loss variation in response to planned energy-restriction among participants from a commercial program and if such variability is related to participant characteristics, including the profile of their gut microbiota. Although subtle, clear differences in gut microbiome signature existed between the weight-loss response groups. Our results suggest contribution of the gastrointestinal microbiota as well as sex and body composition differences toward differential weight-reduction within a retail setting. Additional research—providing independent validation in different cohorts as well as exploring how such knowledge may help customize improved weight-loss approaches—will be key.

## Figures and Tables

**Figure 1 microorganisms-08-01246-f001:**
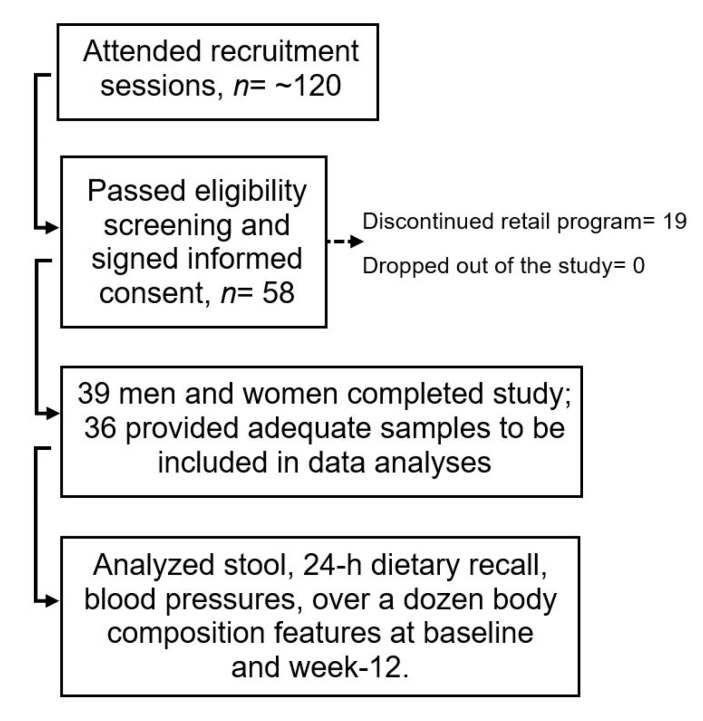
Schematic chart showing participant recruitment in the study.

**Figure 2 microorganisms-08-01246-f002:**
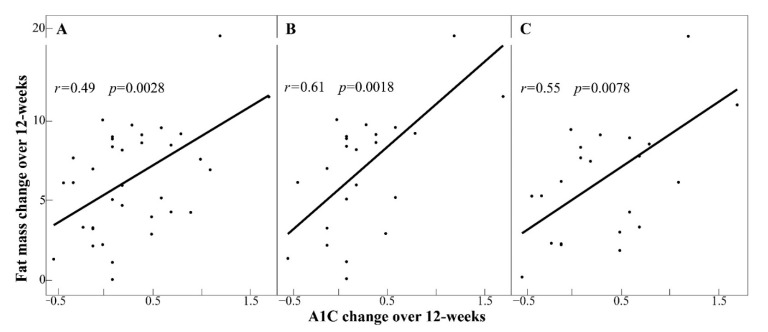
Changes in fat mass and glycated hemoglobin associated in participants with higher BMI and android:gynoid fat. Scatter plots with regression line showing decrease from baseline to Week 12 (*r*, Pearson’s coefficient): (**A**) all participants; (**B**) higher body mass index group (HI_BMI_); and (**C**) higher android:gynoid fat group (HI_AG_). The corresponding low groups for BMI and AG did not show similar association.

**Figure 3 microorganisms-08-01246-f003:**
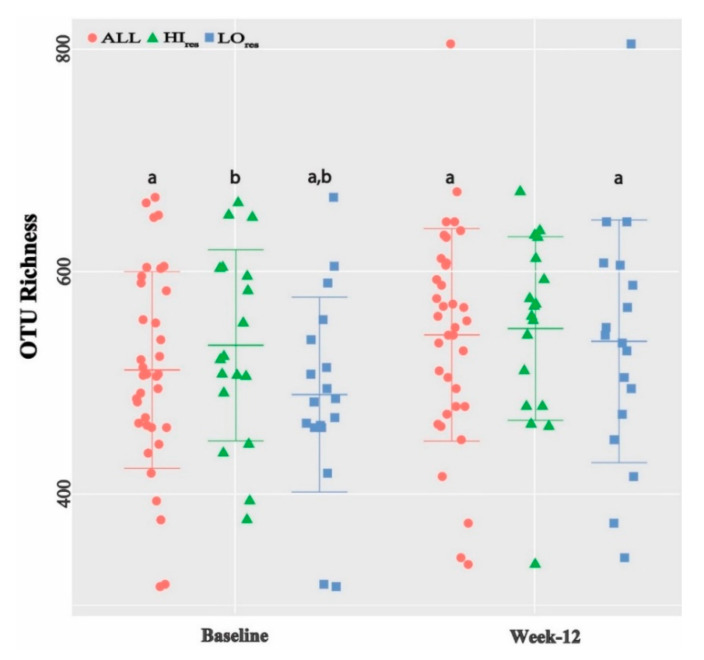
OTU-richness relate with weight-loss response. Data shown as mean ± SE with individual data point distribution. a (within group change) and b (difference between groups at a given time point) indicate *p* < 0.05; HI, high; LO, low; res, response groups.

**Table 1 microorganisms-08-01246-t001:** Average nutrient intakes.

Nutrients	All Participants (ALL, *n* = 36)			High-Response Group (HI_res_, *n* = 18)			Low-Response Group (LO_res_, *n* = 18)		
	Baseline	Week-12	*p*	Baseline	Week-12	*p*	Baseline	Week-12	*p*
Caloric Intake, kcal/d	2327 ± 1163	1818 ± 669	<0.01	2254 ± 1284	1722 ± 699	<0.01	2400 ± 1061	1914 ± 643	<0.01
Protein (%E)	17.9 ± 10.9	24.0 ± 9.1	<0.01	18.8 ± 14.4	26.1 ± 11.5	<0.01	17.0 ± 6.0	21.9 ± 5.3	<0.01
Carbohydrate (%E)	41.7 ± 14.2	38.1 ± 8.8	0.01	44.8 ± 15.5	39.9 ± 9.4	0.01	38.6 ± 12.3	36.3 ± 8.1	0.01
Total Fat (%E)	40.3 ± 12.9	38.3 ± 8.2	NS	38.2 ± 13.6	36.3 ± 9.2	NS	42.5 ± 12.1	40.3 ± 6.7	NS
Fiber, g/d	18.5 ± 9.5	17.8 ± 6.0	NS	19.9 ± 10.8	18.5 ± 7.0	NS	18.1 ± 8.3	17.2 ± 4.8	NS

Data are Mean ± SD based on 24 h dietary recall. Comparison of energy intake between HI_res_ and LO_res_ shows *p* > 0.05 both at baseline and at Week 12 (data not shown in table). %E, percent of total energy; NS, not significant.

**Table 2 microorganisms-08-01246-t002:** Baseline characteristics of study participants including differential features between sub-groups.

	ALL (36)	Age Group (*n*)		Sex Group (*n*)		BMI Group (*n*)		Android:Gynoid Group (*n*)		Weight-Loss Response Group (*n*)	
		YA (20)	OA (16)	Male (9)	Female (27)	HI_BMI_ (24)	LO_BMI_ (12)	HI_AG_ (23)	LO_AG_ (13)	HI_res_ (18)	LO_res_ (18)
Age (year)	45.7 ± 15.8	33.0 ± 6.9 *	61.6 ± 5.9 *	44.9 ± 17.7	46.0 ± 15.4	42.8 ± 15.0	51.5 ± 16.3	50.0 ± 16.1 *	38.0 ± 12.1 *	44.0 ± 16.3	47.4 ± 15.5
Body weight (kg)	101.6 ± 24.5	106.1 ± 24.8	96.0 ± 23.6	115.9 ± 32.5 *	96.9 ± 19.7 *	112.2 ± 23.0 *	80.4 ± 8.1 *	106.9 ± 27.3	92.4 ± 15.3	113.2 ± 27.0 *	90.1 ± 14.8 *
BMI (kg/m^2^)	34.6 ± 7.2	35.1 ± 7.1	33.9 ± 7.5	35.1 ± 8.6	34.4 ± 6.8	38.2 ± 6.0 *	27.3 ± 1.8 *	36.0 ± 7.6	32.1 ± 5.9	37.8 ± 7.6 *	31.4 ± 5.3 *
Hip (cm)	120.3 ± 14.4	123.2 ± 13.0	116.7 ± 15.7	117.2 ± 13.0	121.3 ± 15.0	126.9 ± 12.9 *	107.1 ± 5.4 *	120.0 ± 15.6	120.9 ± 12.7	125.9 ± 14.0 *	114.7 ± 12.8 *
Waist (cm)	112.7 ± 18.2	113.0 ± 18.1	112.2 ± 18.9	119.5 ± 22.6	110.4 ± 16.4	119.9 ± 18.0 *	98.2 ± 5.8 *	118.3 ± 19.2 *	102.7 ± 11.3 *	121.4 ± 17.6 *	103.9 ± 14.5 *
Waist:hip	0.9 ± 0.1	0.9 ± 0.1	1.0 ± 0.1	1.0 ± 0.1 *	0.9 ± 0.1 *	0.9 ± 0.1	0.9 ± 0.04	1.0 ± 0.1 *	0.9 ± 0.1 *	1.0 ± 0.1 ^$^	0.9 ± 0.1 ^$^
Total FM (kg)	41.2 ± 14.4	43.4 ± 14.7	38.4 ± 13.9	39.4 ± 18.7	41.8 ± 13.0	47.3 ± 13.6 *	28.9 ± 5.0 *	42.9 ± 16.3	38.2 ± 10.1	46.7 ± 16.5 *	35.7 ± 9.4 *
Total FFM (kg)	61.1 ± 13.2	63.4 ± 12.9	58.2 ± 13.3	77.1 ± 14.5 *	55.7 ± 7.1 *	65.3 ± 13.5 *	52.5 ± 7.0 *	64.4 ± 14.8 *	55.1 ± 6.7 *	66.9 ± 14.0 *	55.2 ± 9.5 *
Total body fat%	39.7 ± 6.9	40.0 ± 7.1	39.2 ± 6.8	32.5 ± 6.3 *	42.0 ± 5.2 *	41.7 ± 6.7 *	35.6 ± 5.2 *	39.3 ± 7.7	40.3 ± 5.4	40.3 ± 7.6	39.0 ± 6.2
Android FM (kg)	3.7 ± 1.5	3.8 ± 1.6	3.6 ± 1.4	4.1 ± 2.1	3.6 ± 1.3	4.3 ± 1.5 *	2.6 ± 0.6 *	4.1 ± 1.6 *	3.0 ± 0.9 *	4.4 ± 1.7 *	3.0 ± 1.0 *
Android FFM (kg)	5.1 ± 1.3	5.1 ± 1.2	5.1 ± 1.4	6.5 ± 1.5 *	4.6 ± 0.8 *	5.5 ± 1.3 *	4.3 ± 0.7 *	5.5 ± 1.4 *	4.3 ± 0.6 *	5.7 ± 1.5 *	4.5 ± 0.8 *
Android Fat%	8.9 ± 1.3	8.6 ± 1.3	9.3 ± 1.3	10.2 ± 1.2 *	8.5 ± 1.1 *	9.0 ± 1.4	8.8 ± 1.2	9.6 ± 1.1 *	7.8 ± 0.9 *	9.4 ± 1.3 *	8.5 ± 1.3 *
Gynoid FM (kg)	6.7 ± 2.3	7.3 ± 2.3	6.0 ± 2.1	5.9 ± 2.4	7.0 ± 2.3	7.7 ± 2.2 *	4.8 ± 0.9 *	6.6 ± 2.5	7.0 ± 2.0	7.5 ± 2.6 *	6.0 ± 1.8 *
Gynoid FFM (kg)	9.9 ± 2.3	10.5 ± 2.5	9.3 ± 2.0	12.3 ± 2.7 *	9.2 ± 1.6 *	10.8 ± 2.3 *	8.3 ± 1.1 *	10.4 ± 2.6	9.1 ± 1.5	11.0 ± 2.5 *	8.9 ± 1.6 *
Gynoid Fat%	16.4 ± 2.1	17.0 ± 2.2 ^$^	15.8 ± 1.8 ^$^	15.5 ± 2.0	16.7 ± 2.0	16.2 ± 2.2	16.8 ± 1.7	15.4 ± 1.7 *	18.3 ± 1.3 *	16.1 ± 2.1	16.7 ± 2.1
Android:gynoid fat	0.6 ± 0.1	0.5 ± 0.1 ^$^	0.6 ± 0.1 ^$^	0.7 ± 0.1 *	0.5 ± 0.1 *	0.6 ± 0.2	0.5 ± 0.1	0.6 ± 0.1 *	0.4 ± 0.1 *	0.6 ± 0.2 ^$^	0.5 ± 0.1 ^$^
A1C (%)	5.6 ± 0.5	5.4 ± 0.4 *	5.8 ± 0.5 *	5.5 ± 0.3	5.6 ± 0.5	5.6 ± 0.4	5.6 ± 0.5	5.7 ± 0.4 *	5.3 ± 0.4 *	5.6 ± 0.5	5.6 ± 0.5
Systolic BP (mmHg)	139.0 ± 16.4	134.9 ± 16.8 ^$^	144.5 ± 14.5 ^$^	144.9 ± 18.6	137.0 ± 15.4	140.3 ± 17.1	136.5 ± 15.3	145.8 ± 15.2 *	127.5 ± 11.2 *	140.2 ± 17.4	137.9 ± 15.8
Diastolic BP (mmHg)	83.4 ± 9.3	82.9 ± 10.5	84.1 ± 7.7	86.6 ± 13.0	82.3 ± 7.7	84.6 ± 9.9	81.1 ± 7.9	86.7 ± 9.2 *	77.8 ± 6.5*	83.9 ± 11.5	82.9 ± 6.9
Heart rate (bpm)	77.3 ± 14.1	84.6 ± 12.2 *	67.5 ± 10.0 *	78.4 ± 16.2	76.8 ± 13.6	79.4 ± 12.5	73.2 ± 16.5	75.4 ± 14.2	80.4 ± 13.8	79.4 ± 15.4	75.2 ± 12.8

Data are Mean ± SD; *p* ≤ 0.05 (*) and *p* = 0.06–0.08 ($) shown for differences between categorical sub-groups. *n*, number of participants; BMI, body mass index; FM, fat mass; FFM, fat free mass; BP, blood pressure; A1C, glycated hemoglobin as percent of total hemoglobin; YA, young adults; OA, older adults; HI, high; LO, low; res, response.

**Table 3 microorganisms-08-01246-t003:** Characteristics of study participants post weight-loss including differential features between sub-groups.

	ALL (36)	Age Group (*n*)		Sex Group (*n*)		BMI Group (*n*)		Android:Gynoid Fat Group(*n*)		Weight-Loss Response Group(*n*)	
		YA (20)	OA (16)	Male (9)	Female (27)	HI_BMI_ (24)	LO_BMI_ (12)	HI_AG_ (23)	LO_AG_ (13)	HI_res_ (18)	LO_res_ (18)
Age (year)	45.7 ± 15.8	33.0 ± 6.9 *	61.6 ± 5.9 *	44.9 ± 17.7	46.0 ± 15.4	42.8 ± 15.0	51.5 ± 16.3	50.0 ± 16.1 *	38.0 ± 12.1 *	44.0 ± 16.3	47.4 ± 15.5
Body weight (kg)	91.4 ± 22.2	95.7 ± 22.3	86.1 ± 21.5	101.2 ± 29.6	88.2 ± 18.6	100.8 ± 21.0 *	72.7 ± 7.9 *	95.4 ± 24.8	84.4 ± 14.8	98.7 ± 25.6 *	84.2 ± 15.7 *
BMI (kg/m^2^)	31.1 ± 6.6	31.6 ± 6.5	30.5 ± 7.0	30.6 ± 7.4	31.3 ± 6.5	34.3 ± 5.8 *	24.7 ± 1.7 *	32.1 ± 7.1	29.4 ± 5.7	32.9 ± 7.2	29.3 ± 5.7
Hip (cm)	113.0 ± 14.5	115.4 ± 12.4	110.0 ± 16.6	109.7 ± 10.9	114.1 ± 15.5	118.7 ± 14.3 *	101.7 ± 5.2 *	113.2 ± 16.2	112.7 ± 11.3	116.2 ± 14.5	109.9 ± 14.1
Waist (cm)	103.6 ± 16.4	104.6 ± 16.3	102.4 ± 16.9	106.9 ± 20.9	102.5 ± 14.9	109.9 ± 16.0 *	91.0 ± 7.7 *	107.5 ± 18.1 ^$^	96.7 ± 9.9 ^$^	109.7 ± 17.8 *	97.5 ± 12.5 *
Waist:hip	0.9 ± 0.1	0.9 ± 0.1	0.9 ± 0.1	1.0 ± 0.1 *	0.9 ± 0.1 *	0.9 ± 0.1	0.9 ± 0.03	0.9 ± 0.1 *	0.9 ± 0.1*	0.9 ± 0.1*	0.9 ± 0.1*
Total FM (kg)	34.8 ± 13.1	36.7 ± 13.1	32.5 ± 13.3	30.5 ± 15.8	36.3 ± 12.1	40.4 ± 12.2 *	23.7 ± 5.8 *	35.9 ± 14.5	33.0 ± 10.5	37.5 ± 15.3	32.1 ± 10.3
Total FFM (kg)	57.5 ± 12.4	60.0 ± 12.7	54.5 ± 11.8	71.7 ± 14.6 *	52.8 ± 7.1 *	61.3 ± 13.0 *	50.1 ± 6.8 *	60.3 ± 14.3 ^$^	52.6 ± 5.6 ^$^	62.1 ± 13.8 *	53.0 ± 9.2 *
Total body fat%	36.9 ± 7.8	37.2 ± 7.8	36.5 ± 8.0	28.3 ± 6.7 *	39.7 ± 5.9 *	39.3 ± 7.3 *	32.0 ± 6.7 *	36.5 ± 8.5	37.7 ± 6.7	36.6 ± 8.6	37.2 ± 7.2
Android FM (kg)	2.9 ± 13.7	3.0 ± 1.4	2.8 ± 1.4	2.9 ± 1.9	2.9 ± 1.2	3.5 ± 1.4 *	1.9 ± 0.6 *	3.3 ± 1.5 ^$^	2.4 ± 0.8 ^$^	3.3 ± 1.6	2.6 ± 1.1
Android FFM (kg)	4.6 ± 1.2	4.6 ± 1.2	4.6 ± 1.3	5.7 ± 1.7 *	4.3 ± 0.8 *	5.0 ± 1.3 *	3.9 ± 0.8 *	5.0 ± 1.3 *	3.9 ± 0.6 *	5.1 ± 1.5 *	4.2 ± 0.7 *
Android Fat%	8.3 ± 1.3	8.1 ± 1.4	8.5 ± 1.2	9.0 ± 1.6 *	8.0 ± 1.2 *	8.5 ± 1.4	7.9 ± 1.2	8.9 ± 1.1 *	7.1 ± 1.0 *	8.6 ± 1.3	8.0 ± 1.4
Gynoid FM (kg)	5.7 ± 2.2	6.2 ± 2.2	5.1 ± 1.9	4.6 ± 2.2	6.0 ± 2.1	6.5 ± 2.1 *	4.0 ± 0.8*	5.5 ± 2.2	6.0 ± 2.1	6.0 ± 2.4	5.4 ± 1.9
Gynoid FFM (kg)	9.2 ± 2.1	9.6 ± 2.2	8.6 ± 1.9	11.1 ± 2.7 *	8.5 ± 1.5 *	9.9 ± 2.2 *	7.7 ± 1.1 *	9.5 ± 2.4	8.5 ± 1.3	9.9 ± 2.4 *	8.4 ± 1.6 *
Gynoid Fat%	16.4 ± 2.1	16.9 ± 2.2	15.8 ± 2.0	15.7 ± 2.1	16.7 ± 2.1	16.1 ± 2.3	17.1 ± 1.6	15.4 ± 1.7 *	18.2 ± 1.6 *	16.0 ± 2.2	16.8 ± 2.1
Android:gynoid fat	0.5 ± 0.1	0.5 ± 0.1	0.6 ± 0.1	0.6 ± 0.2 *	0.5 ± 0.1 *	0.5 ± 0.1	0.5 ± 0.1	0.6 ± 0.1 *	0.4 ± 0.1 *	0.6 ± 0.1	0.5 ± 0.1
A1C (%)	5.3 ± 0.4	5.2 ± 0.4	5.4 ± 0.5	5.2 ± 0.3	5.3 ± 0.5	5.3 ± 0.4	5.2 ± 0.5	5.4 ± 0.4 *	5.1 ± 0.5 *	5.2 ± 0.4	5.4 ± 0.5
Systolic BP (mmHg)	127.9 ± 18.2	124.6 ± 17.1	132.1 ± 19.2	131.2 ± 14.9	126.8 ± 19.3	130.3 ± 19.4	123.0 ± 15.0	132.3 ± 18.5 *	120.1 ± 15.2 *	129.2 ± 21.3	126.6 ± 15.0
DiastolicBP (mmHg)	75.4 ± 9.7	73.4 ± 9.2	77.9 ± 10.0	78.8 ± 7.2	74.3 ± 10.2	76.1 ± 9.9	73.9 ± 9.5	78.2 ± 8.7*	70.5 ± 9.6 *	76.4 ± 10.1	74.4 ± 9.5
Heart rate (bpm)	74.0 ± 12.4	77.6 ± 11.2 *	69.6 ± 12.7 *	68.6 ± 11.9	75.8 ± 12.2	75.2 ± 11.9	71.6 ± 13.6	73.7 ± 14.4	74.5 ± 8.3	72.8 ± 12.7	75.2 ± 12.3

Data are Mean ± SD; *p* ≤ 0.05 (*) and *p* = 0.06–0.08 ($) shown for differences between categorical sub-groups. *n*, number of participants; BMI, body mass index; FM, fat mass; FFM, fat free mass; BP, blood pressure; A1C, glycated hemoglobin as percent of total hemoglobin; YA, young adults; OA, older adults; HI, high; LO, low; res, response.

**Table 4 microorganisms-08-01246-t004:** Differential changes (% change) within and between subgroups from baseline (pre weight-loss) to Week 12 (post weight-loss).

	1. Male (*p*, pre vs. post)	2. Female (*p*, pre vs. post)	*p,* 1 vs. 2	3. HI_BMI_ (*p*, pre vs. post)	4. LO_BMI_ (*p*, pre vs. post)	*p,* 3 vs. 4	5. HI_AG_ (*p*, pre vs. post)	6. LO_AG_ (*p*, pre vs. post)	*p,* 5 vs. 6	7. HI_res_ (*p*, pre vs. post)	8. LO_res_ (*p*, pre vs. post)	*p,* 7 vs. 8
Body weight	12.9 ± 4.9 (<0.01)	9.0 ± 4.3 (<0.01)	0.03	10.2 ± 5.4 (<0.01)	9.6 ± 2.7 (<0.01)	NS	10.8 ± 4.4 (<0.01)	8.6 ± 5.1 (<0.01)	NS	13.2 ± 3.6 (<0.01)	6.8 ± 3.3 (<0.01)	<0.01
BMI	12.9 ± 4.9 (<0.01)	9.03 ± 4.29 (<0.01)	0.03	10.2 ± 5.44 (<0.01)	9.6 ± 2.7 (<0.01)	NS	10.8 ± 4.36 (<0.01)	8.6 ± 5.1 (<0.01)	NS	13.7 ± 5.0 (<0.01)	6.8 ± 3.3 (<0.01)	<0.01
Hip	6.3 ± 3.3 (<0.01)	5.9 ± 4.2 (<0.01)	NS	6.5 ± 4.4 (<0.01)	5.1 ± 2.6 (<0.01)	NS	5.7 ± 4.1 (<0.01)	6.7 ± 3.6 (<0.01)	NS	7.8 ± 4.0 (<0.01)	4.3 ± 3.1 (<0.01)	<0.01
Waist	10.6 ± 3.8 (<0.01)	7.0 ± 5.3 (<0.01)	0.07	8.1 ± 6.0 (<0.01)	7.4 ± 3.3 (<0.01)	NS	9.1 ± 4.7 (<0.01)	5.7 ± 5.5 (<0.01)	0.06	9.8 ± 4.7 (<0.01)	5.9 ± 5.0 (<0.01)	0.02
Waist:hip	4.6 ± 2.9 (<0.01)	1.0± 5.6 (NS)	0.07	1.6 ± 6.2 (0.09)	2.5 ± 2.8 (0.01)	NS	3.6 ± 3.8 (<0.01)	−1.1 ± 6.2 (NS)	<0.01	2.2 ± 4.9 (0.04)	1.6 ± 5.7 (NS)	NS
Total FM	24.1 ± 9.3 (<0.01)	13.9 ± 7.1 (<0.01)	<0.01	15.1 ± 8.6 (<0.01)	19.1 ± 9.1 (<0.01)	NS	17.4 ± 8.6 (<0.01)	14.8 ± 9.3 (<0.01)	NS	21.4 ± 7.6 (<0.01)	11.5 ± 7.1 (<0.01)	<0.01
Total FFM	7.2 ± 4.7 (<0.01)	5.1 ± 3.8 (<0.01)	NS	6.2 ± 4.7 (<0.01)	4.6 ± 2.1 (<0.01)	NS	6.4± 4.1 (<0.01)	4.4 ± 3.9 (<0.01)	NS	7.4 ± 4.0 (<0.01)	4.0 ± 3.6 (<0.01)	0.01
Body fat%	13.3 ± 7.3 (<0.01)	5.7 ± 4.9 (<0.01)	<0.01	6.1 ± 4.9 (<0.01)	10.8 ± 8.0 (<0.01)	0.04	7.9 ± 6.5 (<0.01)	7.2 ± 6.4 (<0.01)	NS	10.0 ± 6.3 (<0.01)	5.2 ± 5.7 (<0.01)	0.02
Android FM	32.5 ± 16.8 (<0.01)	18.5 ± 8.3 (<0.01)	<0.01	19.4 ± 11.3 (<0.01)	27.3 ± 13.2 (<0.01)	0.07	22.4 ± 12.7 (<0.01)	21.3 ± 12.2 (<0.01)	NS	27.4 ± 13.5 (<0.01)	16.6 ± 8.4 (<0.01)	<0.01
AndroidFFM	12.6 ± 10.0 (<0.01)	7.6 ± 5.4 (<0.01)	0.07	9.3 ± 7.5 (<0.01)	8.0 ± 6.0 (<0.01)	NS	9.4 ± 7.0 (<0.01)	7.9 ± 7.2 (<0.01)	NS	11.5 ± 7.8 (<0.01)	6.2 ± 5.0 (<0.01)	0.02
AndroidFat%	12.2 ± 12.4 (0.02)	5.5 ± 4.8 (<0.01)	0.02	5.3 ± 7.1 (<0.01)	10.8 ± 8.2 (< 0.01)	0.05	6.7 ± 8.3 (<0.01)	8.0 ± 7.0 (<0.01)	NS	8.4 ± 10.1 (<0.01)	5.9 ± 4.4 (<0.01)	NS
Gynoid FM	23.2 ± 10.7 (<0.01)	14.2 ± 7.1 (<0.01)	<0.01	15.9 ± 9.1 (<0.01)	17.5 ± 8.9 (<0.01)	NS	17.2 ± 9.1 (<0.01)	15.1 ± 8.7 (<0.01)	NS	21.8 ± 8.1 (<0.01)	11.1 ± 6.1 (<0.01)	<0.01
Gynoid FFM	10.0 ± 7.4 (<0.01)	6.8 ± 5.5 (<0.01)	NS	8.0 ± 6.6 (<0.01)	6.9 ± 4.9 (<0.01)	NS	8.5 ± 6.1 (<0.01)	6.0 ± 5.8 (<0.01)	NS	9.4 ± 6.0 (<0.01)	5.9 ± 5.7 (<0.01)	0.08
Gynoid Fat%	−1.08 ± 5.16 (NS)	0.3±5.2 (NS)	NS	0.9 ± 4.5 (NS)	−2.1 ± 5.9 (NS)	NS	−0.3 ± 5.7 (NS)	0.2 ± 4.1 (NS)	NS	0.6 ± 4.3 (NS)	−0.7 ± 5.9 (NS)	NS
AG	13.1 ± 11.9 (<0.01)	5.0 ± 7.5 (<0.01)	0.02	4.3 ± 7.9 (0.02)	12.3 ± 9.9 (<0.01)	0.01	6.6 ± 10.2 (<0.01)	7.7 ± 7.7 (<0.01)	NS	7.7 ± 11.3 (0.01)	6.3±7.0 (<0.01)	NS
A1C	6.1 ± 8.6 (0.08)	4.7 ± 8.4 (<0.01)	NS	4.6 ± 8.1 (0.01)	5.9 ± 9.2 (0.04)	NS	5.4 ± 9.3(0.01)	4.4 ± 6.7 (0.04)	NS	6.5 ± 9.0 (<0.01)	3.2 ± 7.5 (0.07)	NS
Systolic BP	8.9 ± 9.1 (0.02)	8.2 ± 11.2 (<0.01)	NS	7.9 ± 9.8 (<0.01)	9.2 ± 12.3 (0.03)	NS	10.0±10.6 (<0.01)	5.7 ± 10.4 (0.07)	NS	9.1 ± 10.6 (<0.01)	7.7 ± 10.8 (<0.01)	NS
Diastolic BP	7.9 ± 10.5 (0.05)	10.5 ± 10.6 (<0.01)	NS	10.6 ± 9.2 (< 0.01)	8.3 ± 12.9 (0.05)	NS	10.2±10.0 (<0.01)	9.3 ± 11.7 (0.02)	NS	9.6 ± 10.7 (<0.01)	10.0 ± 10.7 (<0.01)	NS
Heart rate	11.0 ± 14.1 (0.06)	−0.3 ± 14.6 (NS)	0.05	3.9 ± 14.9 (NS)	0.1 ± 16.1 (NS)	NS	1.3 ± 13.3 (NS)	4.8 ± 18.3 (NS)	NS	6.5 ± 15.1 (0.08)	1.1 ± 14.6 (NS)	NS

Percent change data from baseline are shown as mean ± SD; positive values indicate decrease and negative values indicate increase from baseline to Week 12; *p* ≤ 0.05 (significant) and *p* = 0.06–0.08 (trend) shown, the rest noted as NS (not-significant and no trend). BMI, body mass index; FM, fat mass; FFM, fat free mass; AG, android to gynoid fat ratio; BP, blood pressure; A1C, percentage glycated hemoglobin; HI, high; LO, low; res, response.

**Table 5 microorganisms-08-01246-t005:** Differential microbial abundance related to weight-loss.

Operational Taxonomic Units (OTU)	Log_2_ Fold Change (Standard Error)	Adjusted *p*
(**A**) **OTU change in ALL from baseline to Week 12** * +ve value denotes increase while –ve value denotes decrease at Week 12		
*Parabacteroides distasonis*	1.185 (0.25)	<0.001
*Prevotella 97otu8549*	3.76 (0.93)	0.010
(**B**) **TU change in High-response group from baseline to Week 12 ***		
*Coprococcus 97otu39504*	3.55 (1.00)	0.04
*Prevotella copri*	5.60(1.02)	<0.001
*Prevotella 97otu94784*	5.24 (0.99)	<0.001
(**C**) **OTU change in Low-response group from baseline to Week 12 ***		
*Bacteroides eggerthii*	9.17 (2.12)	0.002
*Prevotella copri*	−3.52 (0.94)	0.014
*Ruminococcus 97otu96826*	4.97 (0.99)	<0.001
*Ruminococcus 97otu31137*	5.91 (1.00)	<0.001
(**D**) **OTU differences between High and Low-response groups at baseline** ** +ve value denotes higher abundance in HI group while −ve value denotes higher abundance in LO group		
*Acidaminococcus* unclassified	4.85 (1.48)	0.030
*Akkermansia muciniphila*	−4.11 (0.85)	<0.001
*Bacteroides eggerthii*	5.72 (1.47)	0.005
*Bacteroides plebeius*	−3.20 (0.97)	0.030
*Dorea* unclassified	−2.20 (0.67)	0.030
*Eubacterium biforme*	−3.73 (0.99)	0.008
(**E**) **OTU differences between High and Low-response groups at Week 12 ****		
*Akkermansia muciniphila*	−3.74 (0.88)	0.003
*Bacteroides uniformis*	−2.59 (0.80)	0.030
*Coprococcus eutactus*	4.11 (1.04)	0.004
*Parabacteroides 97otu73285*	5.18 (1.21)	0.003
*Prevotella 97otu94784*	3.56 (1.02)	0.021

Data show relative abundance of selected bacterial species in each category. Adjusted *p* ≤0.05 were considered significant after false discovery rate correction.

**Table 6 microorganisms-08-01246-t006:** Differential microbial abundance related to age.

Baseline Differences between Age Groups		
Operational Taxonomic Unit (OTU)	Log_2_ Fold change (SE)	Adjusted *p*
*Anaerostipes 97otu18915*	−3.51 (0.99)	0.017
*Bacteroides eggerthii*	−4.78 (1.25)	0.009
*Bacteroides plebeius*	−4.84 (1.18)	0.005
*Eubacterium biforme*	−5.90 (1.06)	<0.001
*Odoribacter* unclassified	5.03 (1.33)	0.009
*Parabacteroides distasonis*	−1.91 (0.60)	0.035
*Paraprevotella 97otu964*	−4.58 (1.37)	0.025
*Prevotella*	−4.03 (1.17)	0.019
*Ruminococcus 97otu31137*	−4.30 (1.05)	0.005
**Week-12 Differences between Age Groups**		
*Acidaminococcus* unclassified	−4.49 (1.31)	0.024
*Bacteroides fragilis*	−3.24 (0.93)	0.022
*Bacteroides plebeius*	−4.60 (1.30)	0.018
*Coprobacillus*	2.02 (0.70)	0.059
*Coprococcus eutactus*	2.47 (0.77)	0.029
*Eubacterium biforme*	−3.30 (1.01)	0.028
*Lachnospira*	−1.77 (0.47)	0.013
*Odoribacter* unclassified	5.91 (1.14)	<0.001
*Oscillospira 97otu16206*	1.52 (0.47)	0.028
*Oscillospira 97otu71987*	2.10 (0.67)	0.040
*Paraprevotella* unclassified	−4.92 (1.05)	<0.001
*Paraprevotella 97otu964*	−4.34 (1.17)	0.011
*Prevotella*	−3.65 (1.21)	0.049
*Roseburia*	−1.92 (0.64)	0.050
*Sutterella 97otu3927*	−4.63 (1.14)	0.004

Data show relative abundance of bacterial species in each category; positive values denote higher abundance in older adults, while negative values denote higher abundance in younger adults. Adjusted *p* ≤ 0.05 were considered significant after false discovery rate correction.

**Table 7 microorganisms-08-01246-t007:** Correlations among bacteria and participant characteristics.

Groups	Pearson’s Coefficient (*r*)
ALL	Body fat%: *Turicibacter* (b, *r* = −0.55, *p* = 0.001), *Christensenella* (b, *r* = −0.61, *p* = 0.01) Weight-loss (change between Week 0 and 12): **Bacteroides eggerthii* (*r* = 0.54, *p* = 0.0007)
High-response group	Weight-loss (change between Week 0 and 12): **Bacteroides eggerthii* (*r* = 0.60, *p* = 0.008)
Older Adults	Glycated hemoglobin: *Anaerotruncus* (pd, *r* = −0.61, *p* = 0.01), *Parabacteroides distasonis* (b, *r* = −0.76, *p* = 0.01) Body fat%: *Christensenella* (b, *r* = −0.76, *p* = 0.01) Android fat%: *Dorea formicigenerans* (pd, *r* = −0.62, *p* = 0.01), *Prevotella* (pd, *r* = −0.63, *p* = 0.01; b, *r* = −0.53, *p* = 0.03), *Eubacterium biforme* (pd, *r* = −0.54, *p* = 0.069; b, *r* = −0.54, *p* = 0.069), *Anaerotruncus* (b, *r* = −0.50, *p* = 0.048)
Young Adults	Body fat%: *Turicibacter* (b, *r* = −0.66, *p* = 0.002), *Ruminococcus torques* (b, *r* = −0.52, *p* = 0.02), *Anaerostipes* (pd, *r* = −0.59, *p* = 0.03)

*r* cut-off ≥0.5, data for *p* ≤ 0.05 are shown only. Positive values indicate inverse correlation. pd, Week 12; b, baseline; * correlation was also observed for BMI.

**Table 8 microorganisms-08-01246-t008:** Piphillin functional interpretation from 16S data.

Name of Pathways from KEGG Database	Baseline % Mean	Week12 % Mean	*p,* Change
path:map01100 Metabolic pathways	16.21	16.31	0.024
path:map01120 Microbial metabolism in diverse environments	4.54	4.62	0.065
path:map02010 ABC transporters	2.82	2.63	0.005
path:map01200 Carbon metabolism	2.58	2.63	0.004
path:map02024 Quorum sensing	1.55	1.47	0.007
path:map00500 Starch and sucrose metabolism	1.32	1.26	0.001
path:map00010 Glycolysis/Gluconeogenesis	1.19	1.21	0.052
path:map00620 Pyruvate metabolism	0.93	0.97	<0.001
path:map00190 Oxidative phosphorylation	0.84	0.88	0.002

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
