# Peer review of "Association of the Gut Microbiota with Weight-Loss Response within a Retail Weight-Management Program"

_microorganisms, 2020, doi:10.3390/microorganisms8081246_

Round 1
Reviewer 1 Report
I read with inteterest this manucript entitled "Association of the gut microbiota with weight-loss response within a retail weight-management program".
This is an observational study and since the gorup of subject is not so large, Authors have to mention this point in the limitations of this study at line 442.
Abstract is not clearly reported " subjects" enrolled: Number and age are necessary. How they have been enrolled? ( i.e. 14 men and 44 women—
80 between ages 20 and 72 years )
The sujbects range is really large,and the subject enrolled are really small. Are they sufficient subject for a good statisitcal analysis? Authors don't report a sample size calculation.
Conclusion at line 458 are not in line con abstract conclusion, I suggest Author to give more details of their findings.
Author Response
Review 1 |
|
|
|
Comments and Suggestions for Authors
- I read with inteterest this manucript entitled "Association of the gut microbiota with weight-loss response within a retail weight-management program". This is an observational study and since the gorup of subject is not so large, Authors have to mention this point in the limitations of this study at line 442.
We are thankful to the reviewer for his/her interest in the presented research. As suggested, we have now discussed our small cohort size under limitation paragraph Line # 457 and 459.
- Abstract is not clearly reported " subjects" enrolled: Number and age are necessary. How they have been enrolled? (i.e. 14 men and 44 women—80 between ages 20 and 72 years )
We apologize for this lapse, which however, was partly due to our efforts to stay within abstract word limit by the journal. We have now added this information on line # 11-13 in the abstract.
- The sujbects range is really large, and the subject enrolled are really small. Are they sufficient subject for a good statisitcal analysis? Authors don't report a sample size calculation.
We believe the subject range in our study reflects real-world observational scenarios. Nonetheless, we are happy to report that we had adequate statistical power to determine weight-loss response. Please also note that the effect size observed was relatively large. We have now clarified this information by mentioning power analysis in line 103.
- Conclusion at line 458 are not in line con abstract conclusion, I suggest Author to give more details of their findings.
We have now added more information to the abstract in line# 24-26. We believe that this change helps better-align the concluding paragraph with the conclusion in the abstract.
Reviewer 2 Report
Authors in presented article decided to invesigate association between retail programs, weight loss and gut microbiota in humans. The subject of presented research is current and can provide insight into correlations between gut microbiota and well-being in humans. The study is well planned, methodology section is thoroughly described, and different aspects of metabolic homeostasis were invesigated. I have a few suggestions and questions related to presented manuscript.
- In introduction section general association between energetic balance, obesity and gut microbiota was quoted. However, authors did not write about potential mechanisms of action of gut microbiota and did not write how gut bacteria can interact with host's organism in the matter of energetic balance. This section might also include 1-2 sentences about association between nutrients, gut bacteria-derived metabolites and weight gain in rats. Recent research reveals that some microbiota-derived metabolites can influence energetic balance and weight gain in mammals.
- When it comes to number of participants included in the study low numer of males stands out while reading the manuscript. It make comparison between males and females more difficult to assess. However, authors explained this aspect of the study in lines 82-84.
- Results presented by authors are interesting, but did not cover the whole area of interactions between gut microbiota and energetic homeostasis. Further research with additional assesment of changes in concentration of gut bacteria-derived metabolites in body fluids would be vital fot better understanding of this topic and would enrich following research projects.
Author Response
Comments and Suggestions for Authors
- Authors in presented article decided to invesigate association between retail programs, weight loss and gut microbiota in humans. The subject of presented research is current and can provide insight into correlations between gut microbiota and well-being in humans. The study is well planned, methodology section is thoroughly described, and different aspects of metabolic homeostasis were invesigated. I have a few suggestions and questions related to presented manuscript.
We thank the reviewer for the kind comments above.
- In introduction section general association between energetic balance, obesity and gut microbiota was quoted. However, authors did not write about potential mechanisms of action of gut microbiota and did not write how gut bacteria can interact with host's organism in the matter of energetic balance. This section might also include 1-2 sentences about association between nutrients, gut bacteria-derived metabolites and weight gain in rats. Recent research reveals that some microbiota-derived metabolites can influence energetic balance and weight gain in mammals.
We have adjusted the introduction section to reflect this suggestion. We have now added several sentences from line # 34-42 as well as 5 new references (#10-14). The added information covers existing research data from mice, rats, and humans on energy balance driven by gut microbiota, especially mediated by nutrient-derived microbial metabolites like SCFA.
- When it comes to number of participants included in the study low numer of males stands out while reading the manuscript. It make comparison between males and females more difficult to assess. However, authors explained this aspect of the study in lines 82-84.
We share the same concern as the reviewer, which is why we tried to clarify it in our original submission in two different sections (methods and Results).
We have noted the following, as the reviewer has noted in his/her comment, under Methods section (line # 89-93):
Volunteers joined the study on a first-come-first serve basis after going through eligibility screening and providing informed consent in writing. Despite our best effort, a low number of males signed up for the study. This experience is similar to another study reporting that men are less likely than women to participate in retail weight-loss programs.30
Further, we have noted the following under Results section (line # 277-279):
Although, the sample size for males was especially smaller in the study, higher weight-loss in males than females align with previous reports from other groups (ref# 56, 57). Furthermore, a previous study reported that men are less likely than women to participate in retail weight-loss programs (30), which are in line with our observation in the presented study.
- Results presented by authors are interesting, but did not cover the whole area of interactions between gut microbiota and energetic homeostasis. Further research with additional assesment of changes in concentration of gut bacteria-derived metabolites in body fluids would be vital fot better understanding of this topic and would enrich following research projects.
We agree with the reviewer. In line # 443-444, we have made a note of this suggestion as one of the possible future direction from the presented research. We have also acknowledged in the limitation paragraph that the scope of this study was exploratory in nature with a relatively smaller (but adequate) cohort size (line #457 and 459).